## RESEARCH ARTICLE

# Transcriptional adaptation after deletion of *Cdc42* in primary T cells

Adam M. Rochussen*, Claire Y. Ma and Gillian M. Griffiths‡,§

## ABSTRACT

*Cdc42* is a Rho family GTPase known for its central role in cell polarity and cytoskeletal regulation. To understand the role of *Cdc42* in polarised secretion from cytotoxic T lymphocytes (CTLs) we used CRISPR/Cas9 gene deletion. Although *Cdc42*-deleted CTLs initially showed reduced cytotoxicity for up to 2 days after CRISPR-mediated deletion, full secretion and cytotoxicity was rapidly restored and even enhanced while CDC42 protein remained absent. In contrast, chemical inhibition of CDC42 using CASIN consistently decreased secretion in wild-type cells, but had no impact on *Cdc42*-deleted CTLs, confirming the specificity of this inhibitor. Comparative proteomics and transcriptomics of CTLs after *Cdc42* deletion revealed transcriptional changes that could support improved T cell function, including compensation via other Rho GTPases. Targeting the promoter region of *Cdc42* did not trigger transcriptional adaptation, consistent with a nonsense-mediated decay mechanism of genetic compensation. Our work highlights the importance of taking orthogonal approaches to study protein function and reveals the remarkable robustness of primary T cells to adapt to loss of an essential gene.

KEY WORDS: Transcriptional adaptation, CRISPR/Cas9, CDC42, T cell, Rho GTPase

## INTRODUCTION

CDC42 is a Ras-like small G protein of the Rho subfamily (i.e. a Rho GTPase). It acts as a regulatory switch (active when bound to GTP and inactive when this GTP is hydrolysed) to control cytoskeletal architecture and polarity across phylogenetically diverse organisms (Baschieri et al., 2014; Johnson and Pringle, 1990; Kodama et al., 1999; Leibfried et al., 2013; Miller et al., 2020; Takai et al., 2001; Zhang et al., 2022). CDC42 is essential during embryogenesis in mice (Chen et al., 2000) and plays an important role in many cell types, including T lymphocytes (T cells), where a crucial role in cell polarity has been postulated (Chemin et al., 2012; Stowers et al., 1995; Tskvitaria-Fuller et al., 2006). The importance of CDC42, and of Rho GTPases more broadly, in T cells is underscored by the plethora of inherited genetic variants that impact immune function (known as human inborn errors of immunity), involving autoinflammation and/or recurrent infections, caused by mutations in genes encoding CDC42, Rac1, RhoA, CDC42 effectors and CDC42 regulators (Bucciol et al., 2020; Burns et al., 2017; Comrie and Lenardo, 2018; Lam et al., 2019; Martinelli et al., 2018; van Wijck et al., 2023).

Here, we set out to determine the importance of CDC42 in cytotoxic T lymphocytes (CTLs), a subset of T cells responsible for the recognition and elimination of cancerous and virally infected cells via polarised secretion of cytolytic proteins at the immune synapse, resulting in target cell lysis. Using CRISPR/Cas9 we found that Δ*Cdc42* CTLs, counter-intuitively, exhibited enhanced secretion and killing. In contrast, chemical inhibition using a CDC42-specific inhibitor consistently reduced CTL-mediated secretion and killing. Using a time course, we found that although *Cdc42* deletion initially decreased CTL killing capacity, this was regained over a period of days. Transcriptomic and proteomic analysis revealed upregulation of genes and proteins in compensatory pathways, leading to 'transcriptional adaptation' (El-Brolosy and Stainier, 2017). We show that targeting the promoter region of *Cdc42* does not trigger transcriptional adaptation, consistent with a nonsense-mediated decay-triggered mechanism put forward by others (El-Brolosy et al., 2019). Finally, we show that potent functional adaptation is specific to CDC42, and not to the other Rho family members Rac1 and RhoA. Our work highlights an indirect effect of CRISPR/Cas9 gene deletion, which seems to affect some genes more than others. We validate alternative approaches to perturb CDC42 function in primary cells, namely specific inhibition and targeting of the promoter region, which can provide the basis for further mechanistic investigation into this key polarity regulator.

## RESULTS

### CRISPR/Cas9-mediated deletion of *Cdc42* results in enhanced CTL function

To target *Cdc42* for CRISPR/Cas9-mediated gene deletion in *ex vivo* murine CTLs, we nucleofected OTI transgenic mouse CTLs with ribonucleoprotein (RNP) complexes containing Cas9 protein, trans-activating CRISPR RNA (tracrRNA) and gene-specific CRISPR RNA (crRNA) targeting three *Cdc42* exons ('Δ*Cdc42*') or a non-targeting sequence ('NT'). After 3–5 days of recovery from nucleofection, we performed functional assays and validated knockout efficiency via western blotting (e.g. Fig. 1J). Unexpectedly, Δ*Cdc42* CTLs exhibited no defect in their ability to kill cancerous targets *in vitro*, killing similarly or even more effectively than NT CTLs (Fig. 1A).

To kill targets, CTLs must migrate and then conjugate with targets via the cytotoxic immune synapse, where they deliver granzymes and perforin via polarised secretion. To test each of these processes, we performed degranulation assays based on the cell surface exposure of LAMP1 (a secretory lysosome membrane

University of Cambridge, Cambridge Institute for Medical Research, Keith Peters Building, Cambridge Biomedical Campus, Hills Road, Cambridge CB2 0XY, UK. *Present address: Salk Institute for Biological Studies, 10010 North Torrey Pines Road, La Jolla, CA 92037, USA. ‡Present address: Yale School of Medicine, Department of Cell Biology, 333 Cedar Street, PO Box 208002, New Haven, CT 06520, USA.

§Author for correspondence (gg305@cam.ac.uk)

A.M.R., 0000-0001-9439-1206; C.Y.M., 0000-0002-4244-7535; G.M.G., 0000-0003-0434-5842

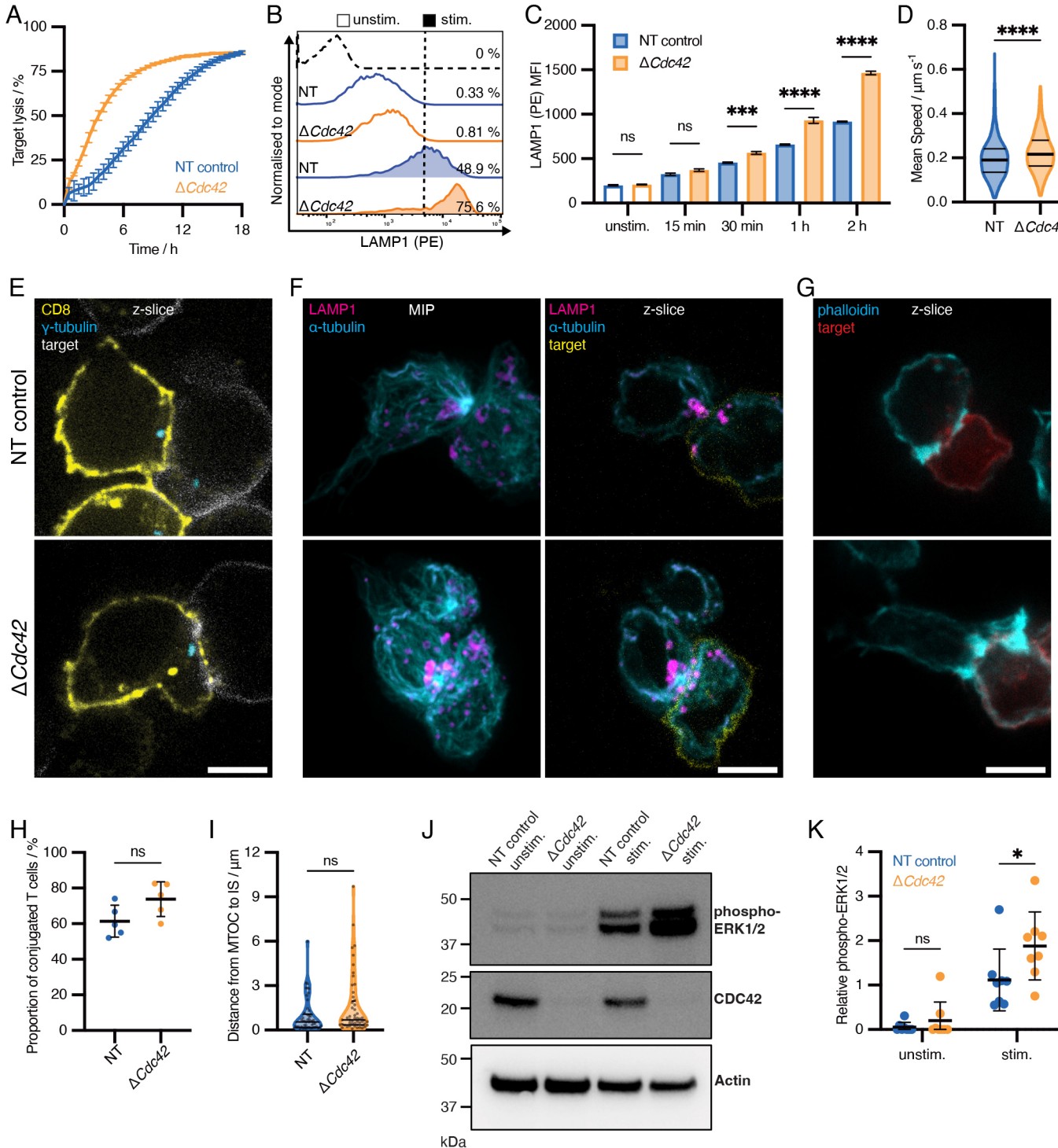

**Fig. 1.** See next page for legend.

protein) and migration assays. Δ*Cdc42* CTLs exhibited increased degranulation (Fig. 1B,C) and migrated more quickly compared to NT CTLs (Fig. 1D). Successful degranulation requires depletion of F-actin at the centre of the immune synapse, polarisation of the microtubule-organising centre (the centrosome in CTLs), and dynein-mediated transport of secretory lysosomes towards the polarised centrosome (Ritter et al., 2015; Stinchcombe et al., 2001, 2006). Using immunofluorescence and confocal microscopy, we confirmed that the rate of conjugate formation, F-actin depletion

across the synapse, centrosome polarisation and secretory lysosome clustering around the centrosome were all comparable between NT controls and Δ*Cdc42* CTLs (Fig. 1E–I).

Underpinning these cell biological processes is signalling downstream of the T cell receptor (TCR). Remarkably, we found that TCR signalling was increased in Δ*Cdc42* CTLs as measured by ERK1 and ERK2 (ERK1/2, also know as MAPK3 and MAPK1, respectively) phosphorylation (Fig. 1J,K). As these data appeared to contrast with results from studies in other immune cell types

**Fig. 1. Improvement of CTL function days after CRISPR/Cas9-mediated loss of *Cdc42*.** (A) IncuCyte killing assay comparing control NT (blue) and Δ*Cdc42* (orange) CTLs 72-h post-nucleofection with CRISPR/Cas9 RNPs. Error bars represent mean±s.e.m. of technical replicates. Representative of 11 independent experiments. (B,C) Degranulation assay based on LAMP1 exposure. (B) Representative histograms showing unstained (dotted curve), NT (blue) or Δ*Cdc42* (orange) OTI CTLs with (filled) or without (unfilled) stimulation. Percentages are 'LAMP1 positive' based on a gate that excludes >99% of the unstimulated NT condition, represented by the vertical dotted line. Representative of five independent experiments. (C) Time-course of degranulation assays (relating to B). Median fluorescence intensities (MFI) are plotted, with three technical replicates per condition. Mean±s.e.m. is shown. Adjusted *P*-values (left to right)=0.9969, 0.1965, 0.0002, <0.0001, <0.0001 (ordinary one-way ANOVA with Šidàk's multiple comparison test). (D) Migration assay. CTLs were pre-stained with CellTracker Green dye and seeded onto ICAM-coated glass. Images were taken every 5 s under a 20× objective lens for a total of 5 min. Cells were masked and tracked over time using TrackMate (ImageJ). Mean track speeds are shown, with each data point representing a cell tracked over time. Data plotted as a violin plot with horizontal lines representing the median and inter-quartile ranges. *P*<0.0001 [unpaired two-tailed *t*-test: NT (mean=0.1958 μm/s, *n*=599) versus Δ*Cdc42* (mean=0.2247 μm/s, *n*=799)]. (E–G) Representative confocal micrographs of NT or Δ*Cdc42* OTI CTLs forming synapses with EL4 target cells expressing farnesyl-mTagBFP2 (E, white; F, yellow; G, red). Cells were fixed after 15 min and then stained with (E) anti-CD8 (yellow, membrane/synapse) and anti-γ-tubulin (cyan, centrosome), (F) anti-LAMP1 (magenta, lysosomes) and anti-α-tubulin (cyan, microtubules), and (G) phalloidin (cyan, F-actin). *Z*-stacks were taken with a spinning disc confocal microscope with slices at 0.15 μm intervals. Single slices are shown for E–G and maximum intensity projections (MIP) additionally shown for F. Scale bars: 5 μm. Representative of at least three biologically independent experiments each with >50 conjugates per condition. (H) Quantification of CTL conjugate formation as a percentage of total T cells per field of view in NT (blue) and Δ*Cdc42* (orange) CTLs (relating to E). Data points represent percentages from a 140 μm×140 μm panel of four acquisitions with an 100x objective lens (20 FOVs total per condition). Graphs show data points and mean±s.d. *P*=0.0709 [two-tailed unpaired *t*-test: NT mean=61.44, Δ*Cdc42* mean=73.76]. Representative of six independent experiments. (I) Analysis of centrosome polarisation of NT (blue) or Δ*Cdc42* (orange) CTLs conjugated with EL4 target cells relating to E. Data points represent 3D distances from centrosome to synapse in individual conjugated CTLs. Horizontal lines represent the medians and quartiles. NT median=0.527 μm, *n*=50; Δ*Cdc42* median=0.6815 μm, *n*=52. *P*=0.0893 (Kolmogorov–Smirnov test). (J,K) Immunoblot analysis of ERK1/2 phosphorylation upon TCR activation. NT or Δ*Cdc42* CTLs were seeded onto stimulatory (pre-coated with anti-CD3ε) or non-stimulatory plates for 40 min. Cells were lysed and phosphorylated ERK1/2 protein content analysed by SDS-PAGE and immunoblotting. (J) Representative immunoblot showing CDC42 to verify CRISPR knockout with actin as a loading control. Molecular mass markers are indicated on the left. Representative of eight independent experiments. (K) Quantification of immunoblot via densitometry. Data was mean-centred for each experiment, with points representing independent experiments (*N*=8). Graphs show data points and mean±s.d. *P*=0.3298 (unstim); *P*=0.0162 (stim.) [paired two-tailed Student's *t*-tests; NT unstim. (mean=0.055) versus Δ*Cdc42* unstim. (mean=0.198); NT stim. (mean=1.115) versus Δ*Cdc42* stim (mean=1.881)]. All experiments performed 72–96 h post-nucleofection with CRISPR/Cas9 RNPs. *P<0.05; ***P<0.001; ****P<0.0001; ns, not significant.

(Bucciol et al., 2020; Chemin et al., 2012; Rossatti et al., 2019; Stowers et al., 1995; Tackenberg et al., 2020; Tskvitaria-Fuller et al., 2006; van Wijck et al., 2023), we sought an alternative method to impair CDC42 function to confirm our results.

## Inhibition of CDC42 with CASIN decreases killing capacity and is highly specific

To perturb CDC42 via an orthogonal method, we used the chemical inhibitor CASIN, which prevents release of GDP from CDC42 and thus locks it in the inactive state (Liu et al., 2018). Contrary to our CRISPR/Cas9 experiments, inhibition of CDC42 resulted

in decreased killing and degranulation compared to the DMSO control, and a slight reduction in migration speed. ERK1/2 phosphorylation, while increased in Δ*Cdc42* CTLs relative to NT control, was unaffected by CDC42 inhibition (Fig. S1C,D). To reconcile our contradictory results, we functionally tested CTLs that had been both targeted by CRISPR/Cas9 and treated with CDC42 inhibitor or DMSO. CDC42-deleted CTLs were unaffected by CASIN treatment across killing, degranulation and migration assays, whereas NT control cells were impaired by CASIN treatment (Fig. 2A–C). These results demonstrate that CASIN is a highly specific inhibitor in CTLs and raised the question of how Δ*Cdc42* CTLs were achieving improved function independently of CDC42.

## CTLs adapt over time to CRISPR/Cas9-induced loss of CDC42

To gain insight into how the cells might be improving functionality after CRISPR/Cas9 nucleofection, we performed a time course of killing assays daily. CDC42 was typically undetectable via western blotting at 1–2 days post-nucleofection, and this population of knockout cells was maintained over time (Fig. 3A; Fig. S1A). Within 1 day of nucleofection, all CTLs, still recovering from electroporation, were unable to kill targets effectively. In the replicate shown, after 66 h, we saw a defect in killing by the Δ*Cdc42* CTLs (Fig. 3B) equivalent to the effect of CASIN treatment (Fig. 2A). This killing defect recovered over time until 138 h post-nucleofection, when the knockout CTLs displayed comparable killing to NT CTLs in this instance (Fig. 3B). Notably, the timing of this adaptation varied between replicates, with most replicates showing improved Δ*Cdc42* CTL function relative to NT CTLs at ~48 h post-nucleofection (Fig. S1B).

## CRISPR/Cas9-mediated loss of *Cdc42* triggers transcriptional adaptation to enhance CTL function

To gain unbiased insight into potential mechanisms for adaptation over time, we performed both bulk RNA sequencing and comparative proteomics of Δ*Cdc42* and NT CTLs after adaptation had occurred. Comparative transcriptomics of six paired replicates revealed 22 significantly [false discovery rate (FDR)<5%] upregulated genes, and one significantly downregulated gene (*Cdc42*) in knockout versus control cells (Fig. 3C). All of the genes encoding proteins of known function that were upregulated in Δ*Cdc42* CTLs can be linked with improved CTL function (Table S1). Upregulated genes could be further categorised into small GTPase-adjacent pathways and general T cell enhancing pathways (Table S1). For example, *Arf2*, encoding the Arf2 small G protein, has been shown to cooperate with Rac1 to enable activation of the WAVE complex (Koronakis et al., 2011), which would directly compensate for loss of CDC42. The gene encoding small GTPase Kras was also upregulated, along with the gene encoding the Kras guanine exchange factor (GEF) Rasgrp2, perhaps explaining the increase in ERK1/2 phosphorylation observed in stimulated Δ*Cdc42* CTLs versus control (Fig. 1J,K). Furthermore, genes encoding RhoA and RhoB GTPases were upregulated, falling marginally above the 5% FDR threshold (Table S1). Outside of small GTPase function, genes encoding proteins relating to vesicle transport and exocytosis (*Drc1* and *Otof*, respectively) were upregulated in Δ*Cdc42* CTLs, which likely play a role in lysosome release at the cytotoxic immune synapse.

Performing comparative proteomics on a different set of samples confirmed that a similar pattern of compensatory changes occurs at both RNA and protein level (Fig. 3D; Table S1). To confirm consistency in the adaptation of Δ*Cdc42* CTLs between samples and to cross-validate our multi-omics data, we correlated the expression of significantly differentially expressed (FDR<10%)

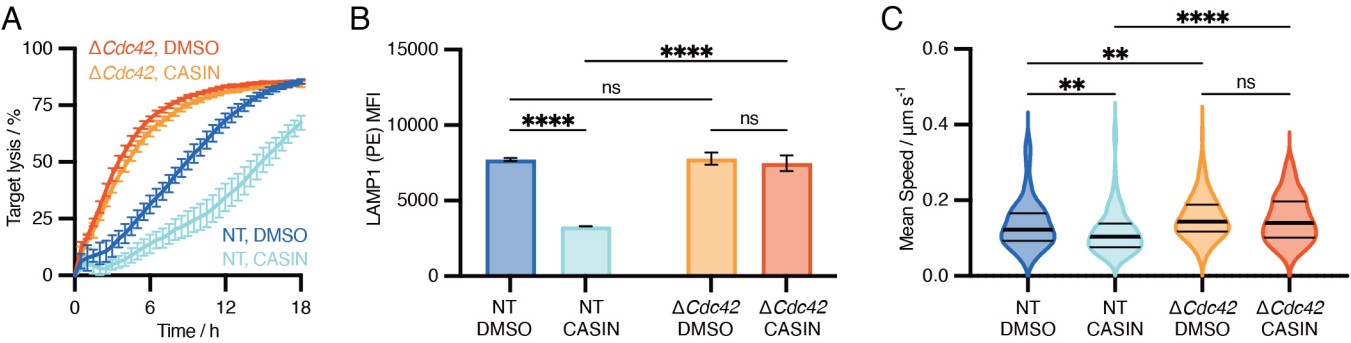

**Fig. 2. CDC42 inhibition does not trigger functional compensation and is highly specific in CTLs.** (A) IncuCyte killing assay comparing NT and ΔCdc42 OTI CTLs treated with either DMSO or CDC42 inhibitor CASIN (5 μM). Representative of three independent experiments. (B) Degranulation assay based on LAMP1 exposure as in Fig. 1. Median fluorescence intensities of stimulated cells combining drug treatment and CRISPR knockout. Results are mean±s.d. From left to right: $P<0.0001$, $P=0.8176$, $P<0.0001$, $P=0.3037$ (two-way ANOVA with Fisher's LSD test). Representative of three independent experiments. (C) Migration assay comparing mean speed for NT and ΔCdc42 CTLs with and without CASIN treatment. Data shown as a violin plot with horizontal lines represent the median and quartiles. NT DMSO=0.125 μm/s, $n=199$; NT CASIN=0.112 μm/s, $n=137$; ΔCdc42 DMSO=0.137 μm/s, $n=173$; ΔCdc42 CASIN=0.132 μm/s, $n=163$. From left to right, $P$-values=0.0056, 0.0068, <0.0001, >0.9999 (Kruskal–Wallis ANOVA with Dunn's multiple comparisons test). Representative of five independent experiments. All experiments performed 72 h post-nucleofection with CRISPR/Cas9 RNPs. $^{**}P<0.01$; $^{****}P<0.0001$; ns, not significant.

genes with the levels of the corresponding proteins, observing a significant positive correlation ($R^2$=0.4919, $P$=0.0006) (Fig. 3E; Fig. S2A). We analysed this combined multi-omics dataset for enriched pathways and protein functions via gene ontology analysis. This revealed that cell polarity, cell adhesion and cytoskeletal organisation were among the significantly enriched pathways (Fig. S2B), and cytoskeletal proteins and small GTPases were among the significantly enriched protein functions (Fig. 3F), consistent with the notion of compensation for loss of *Cdc42*.

### Transcriptional adaptation is specific to targeting of *Cdc42* exons and not its promoter nor other Rho GTPases

Transcriptional adaptation has been described by others across diverse genes, cell types and organisms (Sztal and Stainier, 2020). This phenomenon has been shown to occur in knockouts but not knockdowns and is posited to be mediated by nonsense-mediated decay (NMD) of mutant mRNA (El-Brolosy et al., 2019; Rossi et al., 2015). To test whether this is true in our case, we designed a crRNA targeting the promoter region of mouse *Cdc42* ('ΔPromoter'). Although this produced a more moderate knockout efficiency of *Cdc42* compared to targeting gene exons, we found no evidence of transcriptional adaptation when probing key genes from our multi-omics data via real-time quantitative PCR (RT-qPCR), whereas these were upregulated in ΔCdc42 CTLs (Fig. 4A–D). Degranulation by ΔPromoter CTLs was also slightly reduced relative to NT, whereas ΔCdc42 CTLs exhibited significantly increased degranulation, as above (Fig. 4G,H). These data are consistent with an NMD-mediated mechanism of transcriptional adaptation.

To test whether transcriptional adaptation is specific to *Cdc42* or is generalisable across the Rho GTPase family, we knocked out *Rac1* and *Rhoa* separately by CRISPR/Cas9 (Fig. 4E,F). We found that ΔRac1 and ΔRhoa CTLs exhibited slightly decreased degranulation compared to NT control, in stark contrast to the functionally enhanced ΔCdc42 CTLs. Thus, transcriptional adaptation does not seem to be generalisable across the Rho GTPase family.

### ΔCdc42 CTLs upregulate RhoA and Rac1 activity to compensate for loss of CDC42

There is a high degree of redundancy between Rho GTPases. For example, both CDC42 and Rac1 can regulate Arp2/3-mediated actin

branching via their effectors WASP and WAVE, respectively (Spiering and Hodgson, 2011). To functionally validate the enrichment of Rho GTPase signalling pathways we observed in ΔCdc42 CTLs (Fig. 3E,F; Table S1), we performed G-LISA assays, which assay for GTP-bound (i.e. active) small G proteins. ΔCdc42 CTLs exhibited a significant increase in both Rac1 and RhoA activity compared to NT control (Fig. 5A–C). To test the relevance of this in CTL-mediated killing, we performed killing assays comparing ΔCdc42 and NT CTLs treated with DMSO, CASIN or a Rac1 inhibitor (NSC23766) (Fig. 5D–F). As seen previously, treatment of knockout CTLs with CASIN had minimal impact, whereas NT CTLs were impaired in their killing (Fig. 5E). Conversely, consistent with the notion that ΔCdc42 CTLs undergo transcriptional adaptation that results in a greater reliance on Rac1, inhibition of Rac1 with 100 μM NSC23766 impaired CTL function in ΔCdc42 CTLs more than in NT CTLs (Fig. 5F). Together, our data show that gene deletion of *Cdc42* is partially compensated through increased activity of alternative Rho GTPases that are partially redundant with CDC42, and that this is relevant during CTL-mediated killing.

### DISCUSSION

Our study demonstrates the surprising enhancement of CTL function after CRISPR/Cas9-mediated deletion of *Cdc42*. We show that beneficial transcriptional changes underpin this phenotype in knockout cells, and that it is not observed with either chemical inhibition of CDC42 or CRISPR/Cas9 targeting of the *Cdc42* promoter. The fact that these cells achieve enhanced function, beyond that of non-targeting CRISPR/Cas9 control CTLs, is a striking phenomenon. Such broadscale transcriptional changes across diverse pathways explain this (Fig. 3C–F), but why this happens to such a strong extent in *Cdc42*-deleted CTLs is unclear.

Such genetic compensation after loss of crucial genes has been termed 'transcriptional adaptation' by Stainier et al., and has been reported in *Drosophila*, mice and zebrafish (El-Brolosy and Stainier, 2017; Serobyan et al., 2020; Sztal and Stainier, 2020). This phenomenon involves the activation of compensatory networks to buffer loss of important genes (Rossi et al., 2015; Salanga and Salanga, 2021; Sztal et al., 2018). Mechanistically, the NMD-mediated mechanism proposed by others is caused by the transcription of exons containing indels, which CRISPR/Cas9

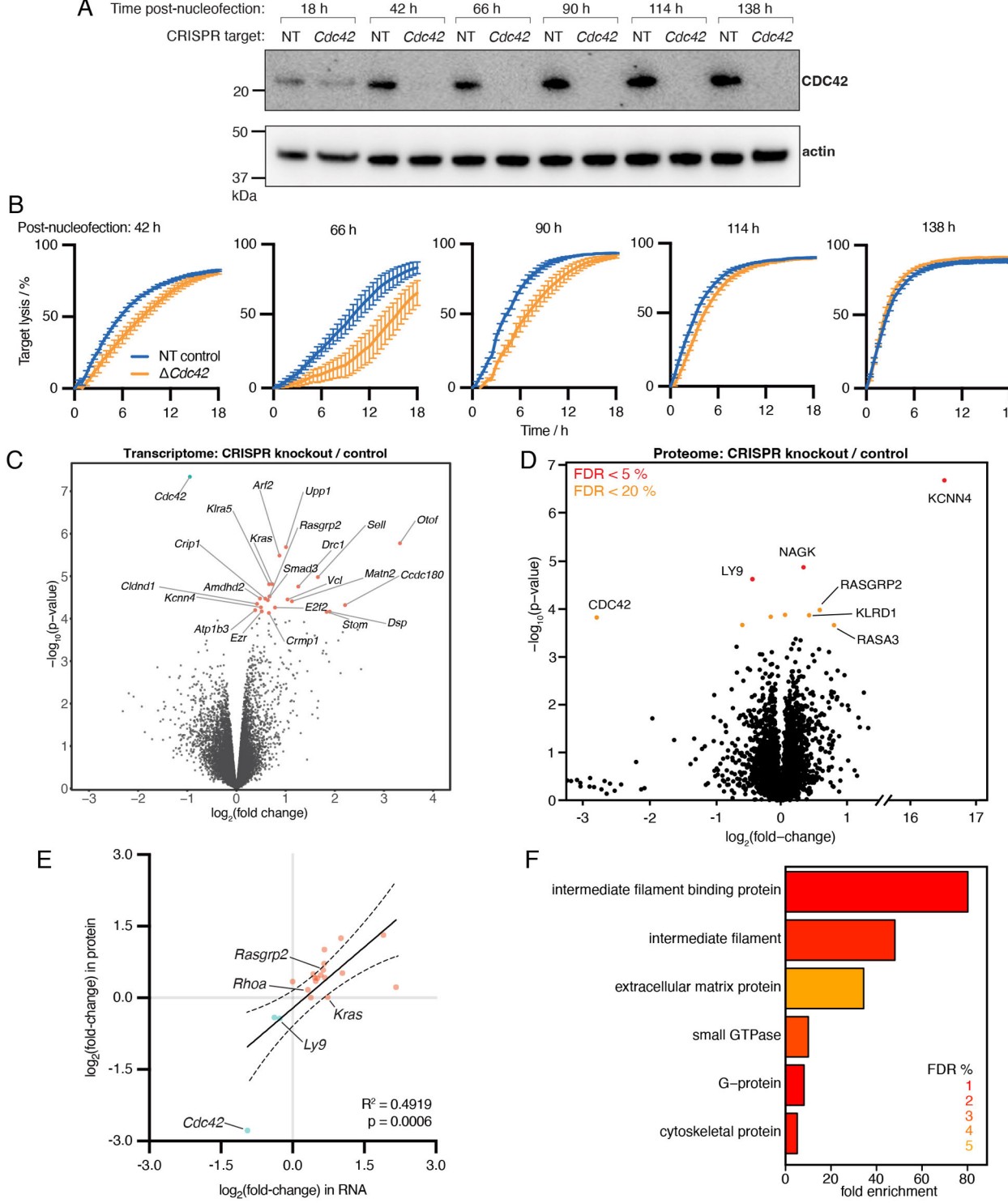

**Fig. 3. CTLs recover function via transcriptional adaptation after loss of *Cdc42*.** (A) Immunoblot showing CDC42 protein levels in NT and Δ*Cdc42* CTLs at daily intervals after nucleofection (performed day 5 post-stimulation). (B) Time course of IncuCyte killing assays at 24 h intervals after nucleofection (relating to A). Representative of three independent time course experiments. Results are mean±s.d. (*n*=6). (C) Comparative transcriptomics by bulk RNA sequencing between six paired replicates of NT and Δ*Cdc42* CTLs 72 h post-nucleofection. Differentially expressed genes with a false discovery rate below 5% are highlighted in red (upregulated) or cyan (downregulated) and labelled. (D) Comparative proteomics by quantitative liquid chromatography and mass spectrometry between four paired replicates of NT and Δ*Cdc42* CTLs 72 h post-nucleofection. Samples are independent to those used for RNA sequencing in C. Differentially expressed proteins with a false discovery rate below 5% are highlighted in red, and below 20% in orange. A number of potentially compensatory proteins are labelled. (E) Correlation between transcriptomic and proteomic changes after CRISPR/Cas9-mediated deletion of *Cdc42*. Genes are shown if they are differentially expressed in either dataset (FDR<10%). Outliers (fold-change>8×) were filtered out and simple linear regression was performed ($y$=0.8534$x$−0.2178, $R^2$=0.4919, $P$=0.0006). The 95% confidence interval is shown with dotted curved lines. Up- (pink) and down-regulated (blue) genes/proteins of interest are labelled. (F) Gene ontology analysis of significantly expressed genes (FDR<10%) performed in PANTHER. Bar plot depicts significantly expressed GO Protein Class terms plotted against fold enrichment, coloured by FDR (%).

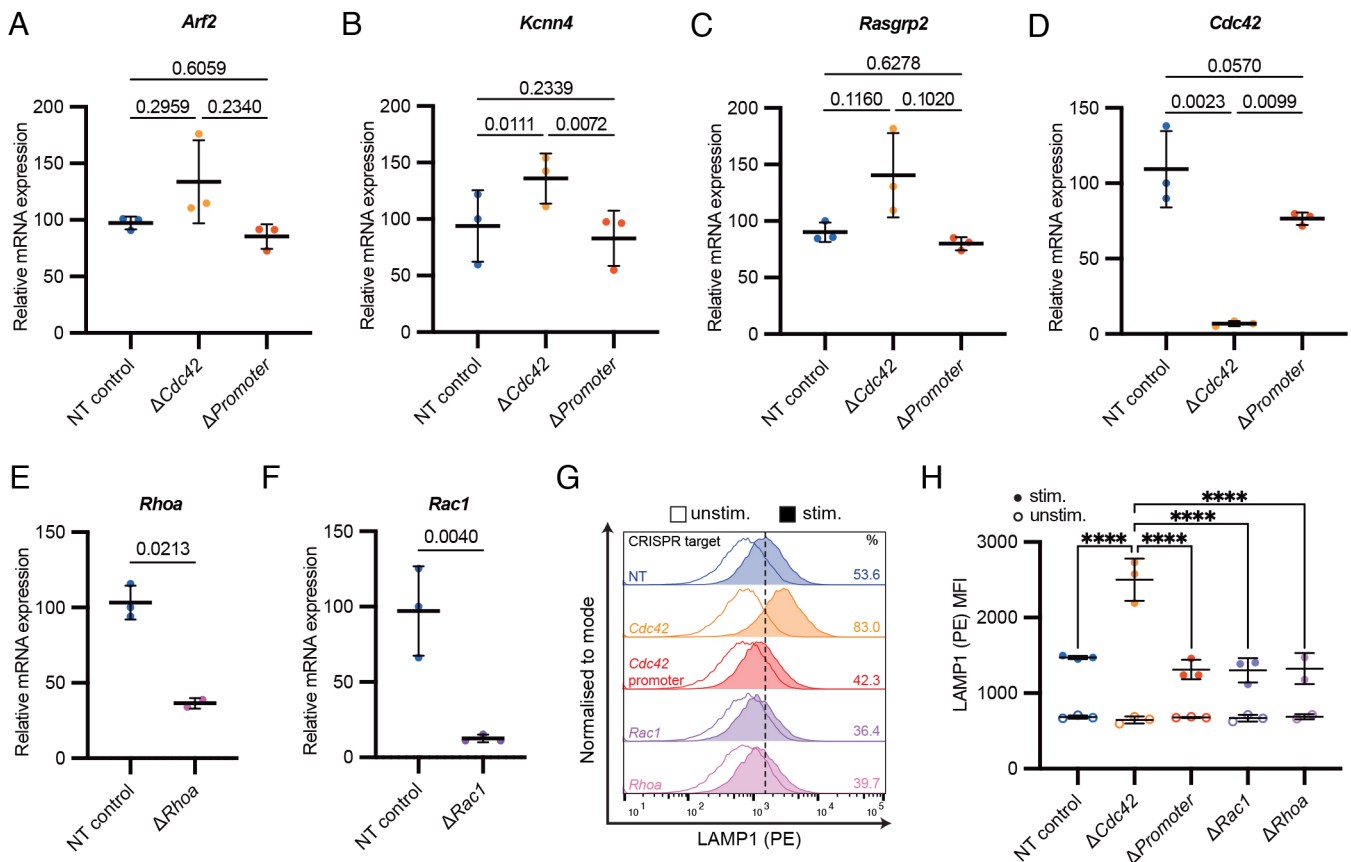

**Fig. 4 . Transcriptional adaptation occurs when targeting *Cdc42* exons, but not the *Cdc42* promoter or other Rho family members.** (A–F) RT-qPCR analysis of mRNA levels between CTLs nucleofected 72 h prior with different CRISPR/Cas9 RNPs. For each sample and mRNA target, expression relative to *Actb* was calculated. This was normalised to one of the NT samples for each mRNA target. Data points are means of technical replicates for biologically independent samples (*n*=3). Mean±s.d. of biological replicates are shown (*N*=3). (A–D) Comparison of *Arf2* (A), *Kcnn4* (B), *Rasgrp2* (C), and *Cdc42* (D) mRNA levels between NT (blue), Δ*Cdc42* (orange), and CTLs nucleofected with RNPs targeting the promoter region of *Cdc42* ('Δ*Promoter*') (red). Samples were compared statistically via repeated measures one-way ANOVA with Holm-Šídák's multiple comparisons test. (E,F) Comparison of *Rhoa* mRNA levels between NT (blue) and Δ*Rhoa* (mauve) CTLs (E) and *Rac1* mRNA levels between NT (blue) and Δ*Rac1* (purple) CTLs (F). Samples were statistically compared via a two-tailed ratio paired *t*-test. (G,H) Degranulation assay based on LAMP1 exposure as in Fig. 1 comparing NT, Δ*Cdc42*, Δ*Promoter*, Δ*Rac1* and Δ*Rhoa* CTLs nucleofected 72 h prior. (G) Representative anti-LAMP1 (PE) histograms. Unstimulated (unfilled) and stimulated (filled) conditions are overlaid for each sample. Percentages denote LAMP1-positive population based on gating at the mode of the unstimulated NT condition, represented by the dashed vertical line. (H) Summary median fluorescence intensity (MFI) data from biologically independent samples (*N*=3). Unstimulated MFI is shown with unfilled circles and stimulated in filled circles. Mean±s.d. of biological replicates are depicted. ****$P$<0.0001 (two-way repeated measures ANOVA with Dunnett's multiple comparisons test). All non-labelled pairwise comparisons are non-significant.

generates (El-Brolosy and Stainier, 2017; El-Brolosy et al., 2019; Sztal and Stainier, 2020). This is consistent with our data, given that NMD would occur after CRISPR/Cas9-generated indels in the *Cdc42* exons that we targeted, whereas chemical inhibition of CDC42 and targeting of the promoter of *Cdc42* would not trigger NMD. Further evidence that NMD-triggering mRNA might be responsible in the case of CDC42 includes the fact that complete gene excision in mice results in embryonic lethality, rather than adaptation (Chen et al., 2000), and human individuals with inborn errors of immunity caused by variants in *CDC42* (which do not trigger NMD) do not exhibit signs of transcriptional adaptation (Martinelli et al., 2018). Future work might isolate the mechanism by selectively inhibiting the NMD kinase SMG1 alongside *Cdc42*-targeting via CRISPR/Cas9, although the effects of SMG1 inhibition alone might be detrimental to primary CTLs (Cook et al., 2024).

Interestingly, we found that functional adaptation was seen exclusively in Δ*Cdc42* CTLs, and not Δ*Rac1* or Δ*Rhoa* CTLs (Fig. 4G,H). Notably, to ensure high efficiency of CRISPR knockout, we targeted three exons of *Cdc42* concurrently, but only one in each

of *Rac1* and *Rhoa*. The abundance of *Cdc42* mRNA is also higher in our cells at baseline than *Rac1* or *Rhoa* mRNA, per our RNA sequencing data (Fig. S2C). Both of these differences could provide an explanation for increased mutant mRNA levels in Δ*Cdc42* CTLs, which would trigger NMD-mediated transcriptional adaptation more strongly. It is also possible that transcriptional adaptation is specific to some genes and not others via an unknown mechanism, which would require further investigation to elucidate.

By directly assaying for Rho GTPase activity and testing susceptibility of CTLs to Rac1 inhibition, we showed that some of the compensation seen in Δ*Cdc42* CTLs is achieved by relying on the partially redundant Rho family members Rac1 and RhoA (Fig. 5). However, redundant proteins might compensate for each other even in the absence of transcriptional adaptation. Indeed, this was likely the case in Δ*Rac1* or Δ*Rhoa* CTLs, which only exhibited mild functional impairment (Fig. 4G,H). The fact that such compensation by redundant proteins does not occur with chemical inhibition of CDC42 is perhaps explained by the difference between the presence of an inactive protein versus the absence of a protein

Journal of Cell Science

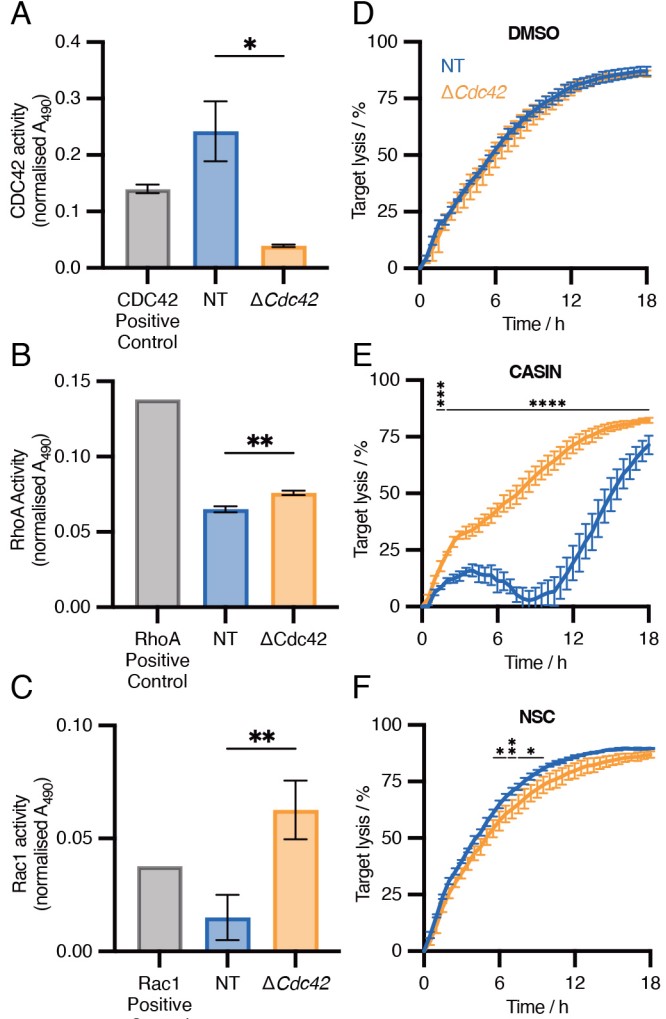

**Fig. 5. Rho family GTPase-adjacent transcriptional adaptation is functionally relevant in CTLs.** (A–C) G-LISA assay for active small G proteins comparing NT and *ΔCdc42* CTLs. Results are mean±s.d. between technical replicates. Unpaired two-tailed *t*-tests performed for each of the three assays. Representative of three independent experiments for each. (A) CDC42 activity, *P*=0.0327; (B) RhoA activity, *P*=0.0073; (C) Rac1 activity, *P*=0.0017. (D–F) IncuCyte killing assays comparing NT and *ΔCdc42* CTLs treated with DMSO (control, A), CASIN (CDC42 inhibitor, 4 μM, B), and NSC23766 (Rac1 inhibitor, 100 μM, C). Concentrations of inhibitors were standardised to twice the IC$_{50}$ for each drug. Results are mean±s.d. (*n*=4). Conditions were compared via two-way ANOVA with Tukey's multiple comparisons test at each time point. Significantly different timepoints are shown. *\*P*<0.05; \*\**P*<0.01; \*\*\**P*<0.001; \*\*\*\**P*<0.0001. Representative of three independent experiments. All experiments performed 72 h post-nucleofection with CRISPR/Cas9 RNPs.

altogether. With CASIN treatment, CDC42-interacting proteins, many of which might interact with other Rho family members at a lower affinity, would still be interacting with inactive CDC42. Meanwhile, deletion of a gene and subsequent removal of a protein from the proteome allows for lower affinity interactions with redundant proteins to occur, enabling greater compensation by family members at the protein level.

Across our experiments, we found variability in the time taken for the population of cells to transcriptionally adapt to loss of *Cdc42* – with improved function in *Cdc42*-deleted cells observed as early as 42 h (Fig. S1B) and as late as 138 h (Fig. 3B) post-nucleofection. Others have reported a high degree of cell-to-cell variability in

NMD efficiency (Sato and Singer, 2021), which might contribute to the spectrum of adaptation timescales we see in our data. In any one experiment, a proportion of cells likely adapts, whereas the rest do not. Over time, the adapted cells could outgrow non-adapted cells. Thus, our readout of population-level adaptation is affected by both the initial proportion of adapted cells (perhaps related to NMD efficiency and nucleofection efficiency), and comparative growth rate of the cells. Cell culture conditions as well as biological differences between mice could affect each of these factors. Thus, future work might focus on later timepoints to ensure a higher proportion of adapted cells within the population, or use techniques with single-cell resolution, such as flow cytometry or single-cell RNA sequencing, to be able to capture any differences in degrees of adaptation within a population.

Taken together, our data highlight an underappreciated cellular response to CRISPR/Cas9-mediated gene deletion in primary CTLs and highlights the need for orthogonal approaches when investigating gene function.

## MATERIALS AND METHODS
### Reagents
Details regarding key reagents are given in Table S4.

### Mice
C57BL/6 (B6)-OTI Rag2$^{-/-}$ (B6.129S6-Rag2tm1Fwa Tg[TcraTcrb]1100Mjb) mice, referred to as OTI mice, were bred and housed in the University of Cambridge Facility. This research has been regulated under the Animals (Scientific Procedures) Act 1986 Amendment Regulations 2012 following ethical review by the University of Cambridge Animal Welfare and Ethical Review Body (AWERB). Mice were killed under Schedule I procedures, and spleens extracted and put in cold PBS by staff at the animal facility. We immediately took spleens and created single-cell suspensions of splenocytes for stimulation with OVA. Spleens from both male and female mice aged 12–30 weeks were used in this study.

### Cell culture
OTI CTLs were generated as previously described (Stinchcombe et al., 2023). Briefly, OTI splenocytes were obtained from freshly harvested OTI mouse spleens, mashed through a 70 μm cell strainer and stimulated with 10 nM OVA$_{257-264}$ peptide (SIINFEKL) (Cambridge Bioscience) in mouse T cell medium [RPMI-1640 medium (Sigma-Aldrich, #1640) supplemented with 10% heat-inactivated FBS (LabTech, #FBS-SA), 50 mM β-mercaptoethanol (Thermo Fisher Scientific, #31350010), 10 U/ml recombinant murine IL-2 (Peprotech, #212-12), 2 mM L-glutamine (Sigma-Aldrich, #G7513), 1 mM sodium pyruvate (Thermo Fisher Scientific, #11360070), and 50 U/ml penicillin and streptomycin (Sigma-Aldrich, #P0781)]. After 3 days, OVA$_{257-264}$ was removed, and cells were thereafter maintained in mouse T cell medium with daily replacement of medium via centrifugation and resuspension. Primary cells were incubated in a humidified atmosphere at 37°C with 8% CO$_2$.

Target cells for OTI CTL imaging experiments were EL4 cells (ATCC: TIB-39; RRID: CVCL_0255) stably expressing farnesyl-5-mTagBFP2 ('EL4-BFP'). Target cells for killing assays were EL4 or P815 cells (ATCC: TIB-64; RRID: CVCL_2154) stably expressing NucLight Red nuclear marker (Sartorius, #4625). Cell lines were maintained in DMEM (Sigma-Aldrich, #D5030) supplemented with 10% heat-inactivated FBS. Cell lines were incubated in a humidified atmosphere at 37°C with 10% CO$_2$. Cell lines were validated for the absence of mycoplasma every 4 weeks of continuous culture.

### Nucleofection with RNP complexes
OTI CTLs were nucleofected with CRISPR/Cas9 RNPs on day 4–6 post-stimulation from splenocytes. 5×10$^6$ CTLs were washed in PBS via centrifugation (200 *g*, 5 min) and resuspension, before final resuspension in 100 μl of nucleofection mix (see below). Cells were electroporated in 100 μl Nucleocuvette™ vessels (Lonza, #V4XP-3024) with a 4D-Nucleofector®×unit (Lonza, #AAF-1003X) using pulse code DN-100.

Journal of Cell Science

CTLs were immediately transferred, with micro-Pasteur pipette, to wells of a six-well plate containing 2 ml pre-warmed nucleofection recovery medium [Ca$^{2+}$-free RPMI (US Biological, #R8999-02A), 5% heat-inactivated FBS, 2 mM L-glutamine, 32 μM 1-thioglycerol (Merck, #M6145), 1.7 mM sodium pyruvate, 20 μM bathocuproine disulfate (Merck, #B1125), supplemented with 1% (v/v) DMSO] prior to use. CTLs were left to recover at 37°C with 8% CO$_2$ for 2–6 h, before addition of 6 ml pre-warmed mouse T cell medium to each well. Cells were cultured normally thereafter by centrifugation (200 $g$, 5 min) and resuspension in fresh medium daily.

### CRISPR/Cas9 RNP generation
Single crRNAs were used for all targets except for *Cdc42*, for which three crRNAs targeting different exons were used in combination to achieve high knockout efficiency. 3.125 μl of 200 μM crRNA was mixed with 3.125 μl of 200 μM tracrRNA (Horizon Discovery, #U-002005-50) in PCR tubes and incubated for 30 min at 37°C. 8.2 μl of 5 μg/μl TrueCut™ Cas9 Protein v2 (Thermo Fisher Scientific, #A36499) was then added, mixed by gentle pipetting, and incubated for a further 15 min at 37°C. To these RNP mixtures, 86 μl P3 primary cell solution was added immediately prior to resuspension of cell pellets for nucleofection. Details for the specific crRNAs used are given in Table S2.

### IncuCyte killing assay
Killing by CTLs over time was measured via loss of target cell nuclei. Cell spheroids were imaged with a IncuCyte S3 Live Cell Analysis System (Sartorius). OTI CTLs and target cells were resuspended in mouse T cell medium at a 1:1 ratio at a density of 5×10$^4$/ml (with drug as appropriate; 30 min pre-incubation was utilised in this instance). 100 μl of CTLs and targets were each added to wells of ultra-low attachment round-bottomed 96-well plates (Corning, #7007). EL4-Nuclight Red target cells were pulsed with 1 μM OVA$_{257-264}$ prior to mixing. P815-Nuclight Red target cells were supplemented with 2 μg/ml αCD3ε (Thermo Fisher Scientific, clone 500A2, #16-0033) prior to mixing. Plates were centrifuged for 5 min at 300 $g$ prior to incubating, with first scan 15–30 min after placing plate in incubator. Plates were scanned every 30 min using a 4× objective lens in both the brightfield and red ($\lambda_{ex}$=567–607 nm, $\lambda_{em}$=622–704 nm) channels. IncuCyte S3 v2018-2021 software (Sartorius) was used to create brightfield and fluorescence masks of each well at each timepoint and calculate the total integrated red fluorescence intensity within the spheroid. Relative target lysis was calculated as:

$$\text{Target lysis}\,(\%) = 100 \times$$
$$\left(1 - \frac{\text{integrated fluorescence intensity at current timepoint}}{\text{integrated fluorescence intensity at first timepoint}}\right).$$

### Degranulation assay
CTLs were resuspended in fresh pre-warmed mouse T cell medium at 10$^6$ cells/ml. For drug treatment, cells were then incubated for 30 mins with DMSO (1:1000) or CASIN (1:1000, 5 μM). Fluorescently conjugated anti-mouse CD107a (LAMP1) (Thermo Fisher Scientific, clone 1D4B, #12-1071) was added to the medium at 2 μg/ml, before seeding CTLs onto flat-bottomed plates that had been pre-coated with 1 μg/ml αCD3ε ('stim.') or PBS ('unstim.'). Plates were incubated at 37°C with 8% CO$_2$ for the indicated time. Plates were then kept on ice and stained with a live-dead marker before analysing with an Attune NxT flow cytometer (Thermo Fisher Scientific).

### Phospho-protein signalling assay
CTLs were resuspended in fresh pre-warmed mouse T cell medium at 10$^6$ cells/ml, with the addition of DMSO or CASIN (1:1000, 5 μM) when applicable followed by a 30 min incubation prior to plating cells. CTLs were seeded onto flat-bottomed plates that had been pre-coated with 1 μg/ml anti-CD3ε ('stim.') or PBS ('unstim.'). Plates were incubated at 37°C with 8% CO$_2$ for 40 min.

For immunoblot analysis of phospho-ERK1/2, plates were transferred to ice and cells resuspended, washed in PBS, then lysed and prepared for SDS-PAGE and immunoblotting (see below).

### Migration assay
CTLs were incubated with 1 μg/ml CellTracker™ Green CMFDA Dye (Thermo Fisher Scientific, #C2925) for 20 min before washing out excess dye via centrifugation and resuspension. CTLs were resuspended at 10$^6$ cells/ml in imaging medium [Phenol Red-free RPMI supplemented with 10% heat-inactivated FBS, 2 mM L-glutamine, 25 mM HEPES (Thermo Fisher Scientific, #15630080), 50 U/ml penicillin-streptomycin. Imaging dishes (Mattek, #P35G-1.5-14-C)] were pre-coated with 0.5 μg/ml ICAM in PBS. 250 μl CTL suspension was added per dish and allowed 30 min at 37°C with 8% CO$_2$ for the cells to adhere and begin crawling. After this time, imaging medium was gently passed over the dish to remove non-adhered CTLs before imaging. Single confocal slices were captured with a 20× objective lens (Leica HC PL APO 20×) in the mid-plane of the cells every 5 s for 5 min per field of view. Three fields of view were taken per condition per experiment.

### Immunofluorescence and immunocytochemistry
Cells were resuspended at 10$^6$ cells/ml in serum-free RPMI and mixed with target EL4 blue cells (pre-incubated with OVA) and then seeded onto coverslips or multi-well slides for conjugate formation. Cells were fixed at the designated time points with −20°C methanol for 5 min on ice, before washing in PBS. Fixed cells were blocked in 1% bovine serum albumin in PBS for 30 min at room temperature, before primary and secondary antibody staining. After staining, cells were mounted with SlowFade™ Glass Soft-set Antifade Mountant (Thermo Fisher Scientific, #S36917) or ProLong™ Glass Antifade Mountant (Thermo Fisher Scientific, #P36984). Confocal microscopy was performed exactly as previously described (Stinchcombe et al., 2023).

### Image analysis
Migration assay data was analysed with ImageJ software (NIH) using the TrackMate plugin. Cells were masked using a Laplacian of Gaussian detector, with an estimated diameter of 11 μm and threshold of 2.0. Migration was tracked between frames via simple LAP tracker, with 5 μm maximum linking distance and zero tolerance for gaps. Tracks with fewer than 10 spots were excluded from analysis. Mean track speed was exported and plotted in Prism software (GraphPad).

Data from imaging of fixed conjugates were analysed and processed for export in Imaris (BitPlane) and/or ImageJ (NIH). Image channels were pseudo-coloured for optimal visual contrast. Distance measurements were performed manually in Imaris software across three dimensions. Data points were transferred to Prism software (GraphPad) for plotting and statistical analysis.

### SDS-PAGE and immunoblotting
Cells to be lysed for analysis by western blotting were counted and washed in cold PBS before centrifugation (400 $g$, 5 min, 4°C). Supernatant was entirely removed. Pellets were either flash frozen on dry ice for storage at −80°C, or placed on ice for immediate lysis. Cells were lysed for 30 min on ice at 2×10$^7$ cells/ml in ice cold western blot lysis buffer [150 mM NaCl, 50 mM Tris-HCl pH 8.0, 1 mM MgCl$_2$, 2% Triton X-100, in ultrapure water with added cOmplete™ protease inhibitor cocktail (Merck, #04693132001)]. Lysates were clarified via centrifugation (10,000 $g$, 10 min). Protein concentration was quantified via Micro BCA™ Protein Assay Kit (Thermo Fisher Scientific, #23235), or equivalent cell numbers were used. Lysates were prepared for SDS-PAGE by addition of NuPAGE™ LDS Sample Buffer (Thermo Fisher Scientific, #NP0007) and NuPAGE™ Sample Reducing Agent (Thermo Fisher Scientific, #NP0004), before boiling (95°C, 5 min). SDS-PAGE was performed using NuPAGE™ 4 to 12%, Bis-Tris, 1.0–1.5 mm Mini Protein Gels (Thermo Fisher Scientific, #NP0335BOX). Proteins were transferred to nitrocellulose membranes using the Trans-Blot® Turbo™ Transfer System (Bio-Rad). Immunoblotting was performed either via manual incubations and washes, or via iBind™ Flex Western Device (Thermo Fisher Scientific, #SLF2000). Secondary antibodies conjugated to horse radish peroxidase were used, chemiluminescence was generated with ECL™ Prime Western Blotting Detection Reagent (Cytiva, #RPN2236), and bands were revealed in a ChemiDoc MP Imaging System (Bio-Rad).

For quantification, densitometry was performed with ImageLab (Bio-Rad) with integrated volume normalised to the loading control for each well. Data across separate experiments were normalised by dividing all values by

the average value across the four conditions. Contrast settings for blots in figure panels were set via the 'auto' feature of ImageLab, which includes mild γ adjustment (0.8–1.2). Full non-adjusted blots are included as Fig. S3.

## Comparative transcriptomics

OTI CTLs from six mice (two female and two male aged 9 weeks 3 days, one male and one female aged 12 weeks and 6 days) were nucleofected with CRISPR/Cas9 RNPs on day 5 post-stimulation. After 72 h, RNA was extracted as previously described (Richard et al., 2023). Briefly, cells were lysed with RLT lysis buffer supplemented with 1% (v/v) β-mercaptoethanol. Lysates were passed through a Qiashredder (Qiagen, #79656). RNA was extracted with RNeasy Plus Mini Kit (Qiagen, #74134), according to the manufacturer's protocol with on-column DNA digestion with RNase-free DNase I (Qiagen #79254). RNA quantity was measured with a Qubit Fluorometer (version 4 Thermo Fisher Scientific) and RNA quality was verified by Bioanalyzer 2100 (Agilent Technologies). Sample order was randomised prior to library preparation using TruSeq Stranded mRNA kit (Illumina) and sequencing was performed on 1 lane of an Illumina NovaSeq6000 at the Cancer Research UK Cambridge Institute (CRUK CI) by a researcher who was not aware of sample treatments.

RNA sequencing analysis was performed as previously described (Richard et al., 2023). Data preprocessing was performed by the CRUK CI bioinformatics core. Briefly, reads were aligned to the mouse genome (annotation from Ensembl GRCm39 version 103) and transcript expression was quantified using Salmon (v1.9.0) (Patro et al., 2017). Samples had between 32 to 59 million mapped exonic reads, equivalent to mapping rates between 95% and 97%. All 12 samples were of good quality and included in downstream analyses.

Differential gene analysis was performed following a published workflow using the Bioconductor package edgeR (v4.2.2) in R (v 4.4.2) (Chen et al., 2016; McCarthy et al., 2012; Robinson et al., 2010). Genes were filtered to remove low-expressed genes. Raw gene counts were transformed and TMM normalised to $log_2$-CPM ($log_2$ counts per million reads). Unsupervised clustering was performed as part of quality control. Based on this, a multifactor design matrix was setup including sample group (gene knock-out versus non-targeting control) and biological replicate, in order to adjust for differences between replicates. Dispersions were estimated and a quasi-likelihood negative binomial generalised log-linear model was fitted to the data. Differential expression between *Cdc42* knockout and non-targeting control was assessed by quasi-likelihood F-test. The false discovery rate was estimated by the Benjamini–Hochberg procedure. Genes were considered differentially expressed when they had an FDR<5%.

## Comparative proteomics

OTI CTLs from four mice (two male and two female aged 17 weeks and 6 days at death) were nucleofected with CRISPR/Cas9 RNPs on day 5 post-stimulation. After 72 h, CTLs were washed twice in cold PBS and dry pellets flash-frozen on dry ice. Sample preparation and mass spectrometry were performed as previously described (Lisci et al., 2021), with an Orbitrap Exploris 480 Mass Spectrometer (Thermo Fisher Scientific). Samples were acquired data-independently (directDIA). Protein copy numbers for each sample were analysed in Excel (Microsoft), including median-centring, calculation of fold-change, paired two-tailed *t*-tests and Benjamini–Hochberg FDR correction. Data was plotted in Excel (Microsoft).

## Cross-validation of multi-omics data and gene ontology analysis

To compare results from transcriptomics and proteomics datasets, we filtered for genes common to both datasets (resulting in a total of 5465 genes) and plotted changes in gene abundance versus changes in protein abundance. To perform correlation analysis, we selected a subset of this dataset, using adjusted *P*-value (FDR) below 10% in either dataset and performed a simple linear regression (after removing outliers). For gene ontology analysis, differentially expressed genes (FDR<10%) from this combined dataset was analysed for enriched pathways via PANTHER (https://www.pantherdb.org/) using the common 5465 genes as the reference gene list (Mi et al., 2019).

## Real-time qPCR

RNA was extracted from cells as for comparative transcriptomics. RT-qPCR was performed via a one-step protocol using pre-designed PrimeTime qPCR

probe assays (Integrated DNA Technologies) according to the manufacturer's protocol as follows (see Table S3 for assay details). Keeping everything on ice and RNase-free, 100 ng of RNA was diluted in a total volume of 4.5 µl of nuclease-free water and added to wells of a 96-well plate (Bio-Rad, #HSP9601) in technical triplicate. Nuclease-free water was used as a no-template control for each probe. Master mixes for each qPCR probe were mixed, consisting of 0.5 µl per well of 20× probe assay and 5 µl per well of PrimeTime™ One-Step RT-qPCR Master Mix (IDT, # 10007065), and then 5.5 µl added to relevant wells. Plates were briefly mixed, centrifuged and sealed, before loading into the CFX96 Touch Real-Time PCR Detection System (Bio-Rad). Reverse transcription was initiated with 15 min at 50°C. Polymerase was activated with 3 min at 95°C. Amplification was then performed with real-time detection of the FAM channel via 40 cycles of 5 s at 95°C and 30 s at 60°C. Mean Cq values from technical replicates were calculated and used as separate data points from biological replicate samples. RNA relative quantity was ascertained by calculating ΔCq of each target gene versus *Actb*, which we validated was unchanged by transcriptional adaptation in our RNA-seq dataset (Fig. S2C). Relative mRNA expression was calculated by $2^{\Delta\Delta Cq}$ relative to an NT control sample.

## G-LISA assays

Colorimetric format GTPase activation assay kits ('G-LISA') for RhoA (Cytoskeleton Inc., #BK124), Rac1 (Cytoskeleton, #BK128), and CDC42 (Cytoskeleton, #BK127) were used according to the manufacturer's protocol.

## Data processing, analysis, and presentation

Data was processed, statistical tests applied and data plotted in Excel (Microsoft) and/or Prism (GraphPad) or R software for transcriptomics. Flow cytometry data was plotted in FlowJo (BD Biosciences). Figures were collated in Illustrator (Adobe). Manuscript was processed in Word (Microsoft).

## Acknowledgements

We thank Mark Bowen and Matthew Gratian at the CIMR microscopy core facility, and Reiner Schulte and Gabriela Grondys-Kotarba at the CIMR flow cytometry core facility for training, access to equipment, and assistance. We thank Doreen Cantrell, Shalini Pathak, and the FingerPrints Proteomics Facility at University of Dundee for proteomics sample preparation and mass spectrometry. We thank Yukako Asano, Jane Stinchcombe, Martin Limbäck-Stokin, Gurpreet Dhaliwal, Maddie Robertson, and Anna Lippert at CIMR for helpful discussions.

## Competing interests

The authors declare no competing or financial interests.

## Author contributions

Conceptualization: G.M.G., A.M.R.; Data curation: C.Y.M.; Formal analysis: A.M.R., C.Y.M.; Funding acquisition: G.M.G.; Investigation: A.M.R., C.Y.M.; Supervision: G.M.G.; Visualization: A.M.R.; Writing – original draft: A.M.R.; Writing – review & editing: G.M.G., A.M.R., C.Y.M.

## Funding

This work was supported by Wellcome Trust grants (217100/Z/19/Z, 102163/B/13/Z and 220543/Z/20/Z). Open Access funding provided by University of Cambridge. Deposited in PMC for immediate release.

## Data and resource availability

RNA-seq data is publicly available: ArrayExpress accession E-MTAB-14519. Code for analysis is available on Zenodo (doi:10.5281/zenodo.15837886) and Github (https://github.com/cym20/Cdc42). The mass spectrometry proteomics data have been deposited to the ProteomeXchange Consortium via the PRIDE partner repository with the dataset identifier PXD063348.

## First Person

This article has an associated First Person interview with the first author of the paper.

## Peer review history

The peer review history is available online at https://journals.biologists.com/jcs/lookup/doi/10.1242/jcs.263826.reviewer-comments.pdf

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
