## [Peer Review File · Journal of Cell Science]

Transcriptional adaptation after deletion of *Cdc42* in primary T cells

Adam M. Rochussen, Claire Y. Ma and Gillian M. Griffiths

DOI: 10.1242/jcs.263826

Editor: Daniel Billadeau

Review timeline

Original submission:	7 January 2025
Editorial decision:	3 February 2025
First revision received:	30 April 2025
Editorial decision:	27 May 2025
Second revision received:	23 June 2025
Accepted:	26 June 2025

Original submission

First decision letter

MS ID#: jcs.263826

MS TITLE: Transcriptional adaptation after deletion of *Cdc42* in primary T cells

AUTHORS: Gillian M Griffiths; Adam M Rochussen; Claire Y Ma

ARTICLE TYPE: Research Article

Dear Dr Griffiths,

We have now reached a decision on the above manuscript.

To see the reviewers' reports and a copy of this decision letter, please go to:

As you will see, the reviewers were intrigued by the data but did raise a number of substantial criticisms that prevent me from accepting the paper at this stage. They suggest, however, that a revised version might prove acceptable, if you can address their concerns. If you think that you can deal satisfactorily with the criticisms on revision, I would be pleased to see a revised manuscript. We would then return it to the reviewers.

Reviewer 1

Advance summary and potential significance to field

The manuscript "Transcriptional adaptation after deletion of *Cdc42* in primary T cells" by Maska Rochussenlenko A. M. et al. demonstrates that the CRISPR/Cas9 deletion of *Cdc42* in primary CTLs promotes transcriptional adaptation to enhance CTL function and upregulates RhoA and Rac1 activity to compensate for the loss of *CDC42*. In contrast, chemical inhibition of *CDC42* using CASIN decreases polarized secretion and CTL activity.

Comments for the author

Although the topic is of interest to the readers of Journal of Cell Science and the manuscript is clear and well-written, I believe that this work requires revision to strengthen its focus and quality.

- To validate the conclusion that CRISPR/Cas9 deletion of Cdc42 in primary CTLs promotes transcriptional adaptation to enhance CTL function, the authors may rescue CDC42 expression in knockout cells and then evaluate CTL activity and transcriptional changes. If this approach does not yield conclusive results, the authors may consider targeting the cdc42 promoter, since it has been reported that alleles that fail to transcribe the mutated gene do not exhibit transcriptional adaptation.
- Additionally, to confirm the short-term effects due to CDC42 inactivation, the authors may use RNA interference to target cdc42 in a specific manner.

Additional minor comments:

- The article mentions that the adaptation time of Cdc42 CTLs varies among replicates. It would be helpful to clarify the factors that may influence this variability.
- Please provide an explanation of the spheroid killing assays (page 8)
- Please provide the quantification for Immunoblot analysis of ERK1/2 phosphorylation in figure 1H

Potential extensions of the study: To gain deeper insight into the mechanism of transcriptional adaptation, it might be useful to perform additional experiments by blocking the nonsense-mediated mRNA decay (NMD) pathway using pharmacological or genetic means to demonstrate the involvement of mRNA decay pathways.

Reviewer 2

Advance summary and potential significance to field

The authors provide evidence of transcriptional adaptation in CTLs engineered to lose a gene essential for the functions that coordinate CTL activation and cytotoxicity. This could be potentially apply to other activation-critical genes.

Comments for the author

In this report the authors investigate the role of the Rho family GTPase CDC42 in mouse CTL-mediated cytotoxicity. Using a CRISPR/Cas9 gene deletion approach, they show that, while CDC42-deficient CTLs show defects in target cell killing early after Cdc42 gene deletion, paradoxically function is not only restored but significantly enhanced at later time points. By contrast, function was impaired in CTLs treated with a chemical inhibitor of CDC42. To gain insights into the enhanced performance of CTLs at late time points after CRISPR/Cas9 Cdc42 deletion, the authors performed transcriptomics and proteomics analyses which revealed modulation in mRNA and protein abundance, with a subset of these alterations potentially related to improved T cell function, including regulators of the actin cytoskeleton. The authors conclude that Cdc42 gene deletion leads to transcriptional adaptation to compensate for the loss of this essential gene.

The notion of transcriptional adaptation underlying the improved performance of CTLs under conditions where function should be compromised as the result of loss of a key regulator of a number of activities that converge on killing, including migration, conjugate formation, signaling and degranulation, is very interesting, particularly as it could be potentially extended to other genes. My main reservation is, however, that the data show indeed a correlation, but not a causal relationship, particularly as no validation has been done on the -omics data.

Point 1. The transcriptomics and proteomics results, highlighting a modulation of T cell function-related genes, is certainly suggestive but not conclusive. Validation by real-time RT-PCR and immunoblot should be done on at least the top candidates. Ideally, the best validated candidates should be also assessed functionally, i.e. the respective genes deleted in Cdc42-deleted CTL to understand whether the improvement in Cdc42-deleted CTL function does not occur when they are lost.

Point 2. Along the same line, at least one other Rho GTPase should be knocked down to test whether adaptation is unique to the loss of CDC42. For example, does Rac1 gene deletion also lead to improved CTL function, and are similar genes modulated? Is the activity of CDC42 enhanced?

Point 3. How do the authors explain the improvement in CTL function at late time points after Cdc42 deletion? One could understand that the CTL defects might be compensated by an upregulation of in the expression of relevant genes, but why should function be further enhanced? This is a point that should be definitely be addressed in the Discussion, which in the present form is far too synthetic and does not dwell much on the data.

Point 4. Regarding the chemical inhibition, why would other Rho GTPases not take over in CTLs when CDC42 function is absent? Again, this point should be elaborated in the Discussion.

Point 5. Figure 1. In relation to the degranulation defects (panel c), is granule convergence around the centrosome impaired in Cdc42-deleted CTLs? Does actin polymerize normally at the CTL contact with the target cell?

Point 6. Figure 3. The time course experiment depicted in panel b shows that cytotoxicity of Cdc42-deleted CTLs is still impaired at 96 hours. Despite the variability highlighted by the authors, would it have not been preferable to use a time point later than 72 hours to test for adaptation? Regarding this figure, the authors state at page 6 "By 138 h post-transfection, the knockout CTLs were better at killing than NT CTLs". This does not match the data, which show a comparable killing (panel b).

First revision

Author response to reviewers' comments

To be pasted into "response to reviewers" box.

Reviewer 1: To validate the conclusion that CRISPR/Cas9 deletion of Cdc42 in primary CTLs promotes transcriptional adaptation to enhance CTL function, the authors may rescue CDC42 expression in knockout cells and then evaluate CTL activity and transcriptional changes. If this approach does not yield conclusive results, the authors may consider targeting the *cdc42* promoter, since it has been reported that alleles that fail to transcribe the mutated gene do not exhibit transcriptional adaptation.

Response: We appreciate the reviewer's suggestions regarding rescue or non-adapting controls. A rescue experiment as described by the reviewer would solidify our conclusions, but it not easily done in primary T cells where multiple electroporation days apart would be required. The idea of targeting the promoter sequence is an excellent one, and we have pursued this avenue instead. As predicted by the reviewer, CRISPR knockout of the promoter region of *Cdc42* did not result in any transcriptional adaptation. Figure 4 now shows RT-qPCR data of some of the top upregulated compensatory genes from our transcriptomics data. Whilst knockout efficiency of *Cdc42* by targeting the promoter was not as efficient as targeting exons of *Cdc42* (fig 4D), it is clear that expression of compensatory genes is not upregulated when targeting the promoter (fig 4A-C). Additionally, functional testing via degranulation assay of CTLs where the promoter has been targeted shows a slight decrease in function, while targeting exons of *Cdc42* leads to marked significant increase in degranulation compared to NT control (fig 4G-H). This adds to our findings significantly and is consistent with an NMD-mediated mechanism for transcriptional adaptation. Lines 171-179 of the results describe and explain these data, and they are further discussed in the second paragraph of the Discussion.

Reviewer 1: Additionally, to confirm the short-term effects due to CDC42 inactivation, the authors may use RNA interference to target *cdc42* in a specific manner.

Response: In addition to targeting the promoter region of *Cdc42*, we attempted RNAi targeting *Cdc42* mRNA. This approach was unsuccessful. Perhaps the time window for successful knockdown is limited in the highly metabolically active CTLs. Nonetheless, we hope the reviewer is satisfied with our data targeting the promoter of *Cdc42* which circumvents transcriptional adaptation and thus provides more mechanistic insight. Given our functional knockout-validation of the inhibitor CASIN (Fig 2), we posit that use of this inhibitor is the most effective way to perturb CDC42 in the short term.

Reviewer 1: The article mentions that the adaptation time of Δ Cdc42 CTLs varies among replicates. It would be helpful to clarify the factors that may influence this variability.

Response: This is an important point to discuss and we have added a paragraph to the Discussion expanding on this (lines 264-278).

Reviewer 1: Please provide an explanation of the spheroid killing assays (page 8)

Response: The “spheroid” killing assay is the same *in vitro* assay using the IncuCyte as in other figures (described in Methods under “IncuCyte killing assay”). We agree that use of the word “spheroid” on page 8 (and nowhere else) is confusing. We have removed this word to hopefully improve clarity.

Reviewer 1: Please provide the quantification for Immunoblot analysis of ERK1/2 phosphorylation in figure 1H

Response: We have done this and included it as figure 1K, using data from across all eight biological replicates and appropriate statistical analysis. Extra details in regards to this analysis have been added to the Methods (lines 455-460).

Reviewer 1: Potential extensions of the study: To gain deeper insight into the mechanism of transcriptional adaptation, it might be useful to perform additional experiments by blocking the nonsense-mediated mRNA decay (NMD) pathway using pharmacological or genetic means to demonstrate the involvement of mRNA decay pathways.

Response: This would be a fascinating extension to the study, but beyond the scope of this paper. We have elaborated on the NMD-mediated mechanism in the second paragraph of the Discussion, including suggestions for future work to block NMD via targeting the NMD kinase SMG1 (lines 233-236).

Reviewer 2: The transcriptomics and proteomics results, highlighting a modulation of T cell function-related genes, is certainly suggestive but not conclusive. Validation by real-time RT-PCR and immunoblot should be done on at least the top candidates. Ideally, the best validated candidates should be also assessed functionally, i.e. the respective genes deleted in *Cdc42*-deleted CTL to understand whether the improvement in *Cdc42*-deleted CTL function does not occur when they are lost.

Response: We agree with the reviewer that the transcriptional changes observed after deletion of *Cdc42* are suggestive of improved T cell function, but not causally proven in all cases. We have performed several additional analyses to validate and corroborate the conclusions from our multi-omics data.

Firstly, to demonstrate the consistency of the transcriptional adaptation phenotype, and the proteomic impact of transcriptional changes, we filtered genes that were common to both datasets and significantly differentially expressed in at least one. This allowed us to perform linear regression between transcriptomic and proteomic data (Fig. 3E). The strong correlation between the two datasets, which use completely independent samples, serves to inter-dependently validate each omics analysis whilst highlighting the consistency of our phenotype. These data are described in lines 154-157.

Secondly, since we chose a select few genes to highlight in the text, we have also included gene ontology analysis (Fig. 3F) to demonstrate which pathways and protein functions are most significantly altered by transcriptional adaptation in an unbiased manner. These data are described in lines 158-163.

Finally, to validate transcriptional changes, per the reviewer's suggestion, we performed RT-qPCR on top compensatory mRNAs from three additional biological replicates, confirming the changes seen in our RNA-seq data (Fig. 4A-C). We additionally included CTLs where the promoter of *Cdc42* was targeted, which would not trigger nonsense-mediated decay, and show no transcriptional adaptation in this case. We hope the reviewer agrees that this validates our data and adds additional mechanistic context to our $\Delta Cdc42$ phenotype. These data are described in lines 171-179.

In terms of additional functional validation, we were unable to create high efficiency double- or triple-knockout primary CTLs. Nonetheless, we believe that we have convincingly shown functional validation for the Rho GTPase-associated adaptations. Figure 5 shows that the function of both Rac1 and RhoA are increased in *Cdc42*-deleted CTLs, as implied by our transcriptomics data. We further validated this by testing susceptibility to Rac1 inhibition in a killing assay (fig. 5F), showing that *Cdc42*-deleted CTLs not only upregulate Rac1 activity relative to control CTLs, but rely more on its activity for killing. We hope that the reviewer appreciates the degree of functional validation we have performed here.

Reviewer 2: Along the same line, at least one other Rho GTPase should be knocked down to test whether adaptation is unique to the loss of CDC42. For example, does Rac1 gene deletion also lead to improved CTL function, and are similar genes modulated? Is the activity of CDC42 enhanced?

Response: Following the reviewer's suggestion, we targeted both Rac1 and RhoA for CRISPR/Cas9-mediated gene deletion in CTLs. In both cases, very high knockout efficiency was achieved (fig. 4E-F). Both $\Delta Rac1$ and $\Delta RhoA$ CTLs exhibited slightly reduced function relative to NT control CTLs (fig. 4G, H), while targeting *Cdc42* in the same experiment resulted in significantly increased CTL function. Therefore, we suggest that transcriptional adaptation is somewhat specific to *Cdc42* in this context. A more marked decrease in function of $\Delta Rac1$ and $\Delta RhoA$ CTLs is likely masked by redundancy between these two Rho GTPases, which overlap in function to a greater extent than either does with CDC42. These data are described in lines 181-186. We expand upon these observations in the discussion (lines 238-247).

Reviewer 2: How do the authors explain the improvement in CTL function at late time points after *Cdc42* deletion? One could understand that the CTL defects might be compensated by an upregulation of in the expression of relevant genes, but why should function be further enhanced? This is a point that should be definitely be addressed in the Discussion, which in the present form is far too synthetic and does not dwell much on the data.

Response: It is unclear to us why function is further enhanced beyond that of control cells in our experiments. Transcriptional adaptation seems to be a broadly-acting powerful phenomenon, and such a response would not presumably be able to tune itself to match the initial function of the CTLs prior to gene deletion. This is an interesting and enduring conundrum which we have highlighted in the Discussion (lines 213-216).

Reviewer 2: Regarding the chemical inhibition, why would other Rho GTPases not take over in CTLs when CDC42 function is absent? Again, this point should be elaborated in the Discussion.

Response: The reviewer raises an important question that we did not sufficiently settle in our Discussion. We have amended the Discussion now to include a reflection on the differences between the nature of inhibition and gene deletion, and relevance to compensation by other redundant proteins (lines 254-262).

Reviewer 2: Figure 1. In relation to the degranulation defects (panel c), is granule convergence around the centrosome impaired in *Cdc42*-deleted CTLs? Does actin polymerize normally at the CTL contact with the target cell?

Response: Since *Cdc42*-deleted CTLs degranulate to a greater extent than control CTLs, we did not expect any changes in granule convergence around the centrosome or actin dynamics at the immune synapse. We have now confirmed this via immunofluorescence, observing intact granule convergence, F-actin accumulation around the perimeter of the synapse, and F-actin depletion

across the centre of the synapse in $\Delta Cdc42$ as expected (Fig. 1F, G). These data are contextualised and described on lines 87-95.

Reviewer 2: Figure 3. The time course experiment depicted in panel b shows that cytotoxicity of Cdc42-deleted CTLs is still impaired at 96 hours. Despite the variability highlighted by the authors, would it have not been preferable to use a time point later than 72 hours to test for adaptation? Regarding this figure, the authors state at page 6 "By 138 h post-transfection, the knockout CTLs were better at killing than NT CTLs". This does not match the data, which show a comparable killing (panel b).

Response: We appreciate the point raised by the reviewer here and have added to the discussion to reflect on the fact that larger adaptive changes to the transcriptome and proteome may be observed at later timepoints (lines 274-278). Our discussion of the variability in timescales of transcriptional adaptation has been significantly deepened in general (lines 264-278). Regarding the choice of 72 hours post-nucleofection, this was sufficient to see functional adaptation in almost all cases and was sufficient to see significant changes to mRNA and protein levels. Several of our many independent replicates (e.g. N= 11 NT vs $\Delta Cdc42$ killing assays) were performed at 96 hours post-nucleofection, yet we saw strong compensation at both timepoints. We have amended the sentence referred to by the reviewer to reflect the fact that our data shows comparable killing by 138 hours in this example.

Second decision letter

MS ID#: jcs.263826R1

MS TITLE: Transcriptional adaptation after deletion of Cdc42 in primary T cells

AUTHORS: Gillian M Griffiths; Adam M Rochussen; Claire Y Ma

ARTICLE TYPE: Research Article

Dear Dr Griffiths,

We have now reached a decision on the above manuscript.

To see the reviewers' reports and a copy of this decision letter, please go to:

As you will see, one reviewer has asked for one minor experiment and a quantitation of immunoblots.

Reviewer 1

Advance summary and potential significance to field

The study shows that knocking out Cdc42, a key gene involved in CTL activation and cytotoxicity, triggers a compensatory transcriptional response that enhances CTL function. These findings suggest that similar transcriptional adaptation could occur with other genes critical for CTL activation.

Comments for the author

I have reviewed the revised version of the manuscript titled "Transcriptional adaptation after deletion of Cdc42 in primary T cells". I appreciate the author responses to the comments raised by the reviewers. The manuscript has been significantly improved in both clarity and content. I believe the current version meets the standards for publication and provides a valuable contribution to the field. I recommend acceptance without further revisions.

Reviewer 2*Advance summary and potential significance to field*

The authors provide evidence of transcriptional adaptation in CTLs engineered to lose a gene essential for the functions that coordinate CTL activation and cytotoxicity. This could be potentially apply to other activation-critical genes.

Comments for the author

The authors have addressed satisfactorily all the issues raised in my previous review, providing new data, new analyses and substantially restructuring the manuscript. I have two minor issues that I feel should be addressed:

1. Paragraph lines 97-103: The authors show that TCR signaling was increased in deltaCdc42 CTLs, using ERK1/2 phosphorylation as readout (Fig.1J,K). Because of this (and of the data presented in the previous paragraph), which appeared in contrast with published reports, they used a chemical inhibitor of CDC42 as an alternative method to perturb it. It would be appropriate to test the effect of CASIN on ERK1/2 phosphorylation, which would be expected to be reduced.
2. Lines 200-203: The authors state that "inhibition of Rac1 with 100 microM NSC23766 impaired CTL function in deltaCdc42 CTLs more than in NT CTLs". This is not very evident from the plots shown in figure 5F. These data should be quantified (with stats).

Second revisionAuthor response to reviewers' comments

1. Paragraph lines 97-103: The authors show that TCR signaling was increased in deltaCdc42 CTLs, using ERK1/2 phosphorylation as readout (Fig.1J,K). Because of this (and of the data presented in the previous paragraph), which appeared in contrast with published reports, they used a chemical inhibitor of CDC42 as an alternative method to perturb it. It would be appropriate to test the effect of CASIN on ERK1/2 phosphorylation, which would be expected to be reduced.

We thank the reviewer for this sensible suggestion. We have performed the assay as described. We found no impact of CASIN on ERK1/2 phosphorylation across three biologically independent assays. These new data are shown in Fig. S1C and D and described in lines 110-111. We believe that CDC42 may not play a direct role in TCR signalling in CTLs, and the increased phosphor-ERK signalling seen in knockout CTLs is likely caused by increased Kras activity (upstream in the MAPK pathway) resulting from transcriptional adaptation (as shown in Fig. 3C and 3E). We have included this explanation for our data in the text (lines 147-150).

2. Lines 200-203: The authors state that "inhibition of Rac1 with 100 microM NSC23766 impaired CTL function in deltaCdc42 CTLs more than in NT CTLs". This is not very evident from the plots shown in figure 5F. These data should be quantified (with stats).

Since the difference between NSC-treated CTLs is small, we agree with the reviewer that it is important to show statistical comparisons. We have performed two-way ANOVA with multiple comparisons at each time point between conditions in figure 5D-F. Significantly different timepoints are denoted with asterisks. Details of the tests are described in the figure caption (lines 813-816).

Third decision letter

MS ID#: jcs.263826R2

MS Title: Transcriptional adaptation after deletion of Cdc42 in primary T cells

Authors: Gillian M Griffiths; Adam M Rochussen; Claire Y Ma

Article Type: Research Article

Dear Dr Griffiths,

I am happy to tell you that your manuscript has been accepted for publication in Journal of Cell Science, pending standard publication integrity checks.